# Optimizing Messenger RNA Analysis Using Ultra-Wide Pore Size Exclusion Chromatography Columns

**DOI:** 10.3390/ijms25116254

**Published:** 2024-06-06

**Authors:** Valentina D’Atri, Honorine Lardeux, Alexandre Goyon, Mateusz Imiołek, Szabolcs Fekete, Matthew Lauber, Kelly Zhang, Davy Guillarme

**Affiliations:** 1School of Pharmaceutical Sciences, University of Geneva, CMU—Rue Michel Servet 1, 1211 Geneva, Switzerland; valentina.datri@unige.ch (V.D.); honorine.lardeux@unige.ch (H.L.); 2Institute of Pharmaceutical Sciences of Western Switzerland, University of Geneva, CMU—Rue Michel Servet 1, 1211 Geneva, Switzerland; 3Synthetic Molecule Analytical Chemistry, Genentech Inc., 1 DNA Way, South San Francisco, CA 94080, USA; goyon.alexandre@gene.com (A.G.); zhang.kelly@gene.com (K.Z.); 4Waters Corporation, 1211 Geneva, Switzerland; mateusz_imiolek@waters.com (M.I.); szabolcs_fekete@waters.com (S.F.); 5Waters Corporation, Milford, MA 01757, USA; matthew_lauber@waters.com

**Keywords:** size exclusion chromatography, messenger RNA, ultra-wide pore, aggregates, fragments

## Abstract

Biopharmaceutical products, in particular messenger ribonucleic acid (mRNA), have the potential to dramatically improve the quality of life for patients suffering from respiratory and infectious diseases, rare genetic disorders, and cancer. However, the quality and safety of such products are particularly critical for patients and require close scrutiny. Key product-related impurities, such as fragments and aggregates, among others, can significantly reduce the efficacy of mRNA therapies. In the present work, the possibilities offered by size exclusion chromatography (SEC) for the characterization of mRNA samples were explored using state-of-the-art ultra-wide pore columns with average pore diameters of 1000 and 2500 Å. Our investigation shows that a column with 1000 Å pores proved to be optimal for the analysis of mRNA products, whatever the size between 500 and 5000 nucleotides (nt). We also studied the influence of mobile phase composition and found that the addition of 10 mM magnesium chloride (MgCl_2_) can be beneficial in improving the resolution and recovery of large size variants for some mRNA samples. We demonstrate that caution should be exercised when increasing column length or decreasing the flow rate. While these adjustments slightly improve resolution, they also lead to an apparent increase in the amount of low-molecular-weight species (LMWS) and monomer peak tailing, which can be attributed to the prolonged residence time inside the column. Finally, our optimal SEC method has been successfully applied to a wide range of mRNA products, ranging from 1000 to 4500 nt in length, as well as mRNA from different suppliers and stressed/unstressed samples.

## 1. Introduction

The 2020 COVID-19 vaccine raised public awareness of messenger RNA (mRNA) technology, which in turn brought significant resources to the field to accelerate research and development. However, vaccinating against a pathogen is not the only clinical application of mRNA. Scientists have been exploring mRNA for decades (it was discovered 60 years ago), trying to unlock its potential to prevent and treat diseases. Karikó and Weissman were awarded a Nobel Prize for their breakthrough discovery on utilizing chemically modified mRNA bases to suppress the body inflammation response and in turns enable the therapeutic application of modified mRNA [1]. The main advantage of mRNA is that the encoded proteins are produced inside the cell, so they can directly engage intracellular targets, making them more effective than injected proteins, which can only work through extracellular interactions [2]. Current mRNA-based drugs in development can be divided into three main applications based on their underlying mechanisms of action [3]. This includes prophylactic vaccines, therapeutic vaccines, and therapeutic drugs.

According to industry statistics, there are several hundred mRNA drugs in the clinical pipeline worldwide, with R&D mainly focused on vaccines, which account for about 84%, while therapeutic drugs make up the remaining 16% [4]. Compared to traditional small molecules and antibodies, mRNA drugs can be considered to be a disruptive therapeutic technology [5], with the advantages of having a modular design, short development cycle, strong target specificity, broad therapeutic range, rapid production methods, and good overall tolerability. Clinical vaccine batches can be produced within weeks of the availability of the sequence encoding an immunogen. The production process is cell-free and scalable [6]. mRNA drugs and vaccines are now being widely evaluated for the treatment of rare genetic diseases, several types of cancers, and respiratory and infectious diseases, and they are expected to soon become one of the major class of therapeutic products after small molecules, antibody drugs, and peptides [4,7].

Based on all of these advantageous features, the scientific community is increasing its efforts in the research and development of mRNA drugs. However, as the successful transmission of this technology has been relatively recent, there is a clear need to develop improved analytical workflows that can assess the quality and safety of these complex mRNA-based drug substances and products [8]. Indeed, there are numerous critical quality attributes (CQAs) that need to be evaluated to determine the quality of mRNA drugs. These include mRNA identity (sequence confirmation), integrity (% intact/fragment mRNA + mRNA integrity + 5′ cap residue + 3′ poly(A) tail), content (mRNA concentration), and purity (process- and shelf life-related impurities) [9,10].

The evaluation of mRNA integrity comprises the critical assessment of impurities such as fragments and aggregates, resulting from either the generation of mRNA by in vitro transcription (IVT) from a DNA template or its subsequent purification, formulation, and storage [11]. This is commonly assessed by capillary gel electrophoresis (CGE) using denaturing conditions and has been shown to be applicable for RNA products containing more than 2000 nt. However, capillary electrophoresis (CE)-based RNA integrity assays require relatively long run times and can sometimes show relatively high quantitation variability [8]. Microchip capillary electrophoresis (mCE) has recently been proposed as a quicker alternative for RNA samples consisting of up to 6000 nt [12,13]. Alternatively, ion-pairing reversed-phase liquid chromatography (IP-RPLC) has been widely used for the analysis of small oligonucleotides (ONs), but its application to larger ONs such as mRNA drugs is much more limited. To date, only a few applications have reported the use of IP-RPLC for mRNA using triethylammonium acetate (TEAA) as the main mobile phase additive, as well as polystyrene-divinylbenzene (PS-DVB) polymeric stationary phases with very large pore sizes ranging from 1000 to 4000 Å [14]. Alternatively, a patent application (now abandoned) from Moderna that was dedicated to the characterization of mRNA stability and heterogeneity reported the use of a C18 bonded 300 Å hybrid silica column [15]. Finally, a study demonstrated in 2021 the successful use of another wide pore DVB polymeric column [16]. Each of these approaches has shown the ability to successfully analyze large RNAs of up to 6000–8000 nt in length.

The potential of SEC to characterize mRNA samples has also been studied recently, with the advantage that such separations can be performed under non-denaturing conditions. Goyon et al. were the first to demonstrate the measurement of mRNA aggregates using a prototype ultra-wide pore SEC column and highlighted the possibility to establish the nature of aggregates (covalent vs. non-covalent) via a heat treatment experiment [17]. Camperi et al. performed a comparison of CGE, mCE, and SEC for the evaluation of covalent and non-covalent aggregates of mRNA under native and denatured conditions. Their results, obtained with a commercial ultra-wide pore SEC column packed with 1000 Å particles, confirm the possible application of SEC to assess aggregation of three different mRNA samples (i.e., eGFP, Fluc, and beta gal mRNA) from two different suppliers [13]. More recently, De Vos et al. investigated the capabilities and limitations of SEC for the characterization of IVT mRNA samples of different sizes [18]. The effect of mobile phase composition (ionic strength and organic modifier), pH, column temperature, and pore size (300 Å, 1000 Å, and 2000 Å) on the separation performance and structural integrity of IVT mRNA was described. The authors observed that the resolution of aggregates can be enhanced at elevated temperatures (up to 80 °C), but this approach is only recommended for characterization of aggregates and not fragments since pronounced degradation of the samples were observed under such conditions. Prudent selection of SEC column temperatures might also be needed to ensure that sufficiently long column lifetimes can be achieved wherever bonded packing materials are used.

The aim of the present study was to contribute to the continued development of the mRNA analytical toolbox, with the idea of finding generic SEC conditions that might enable the analysis of a wide range of mRNA products. In a first step, we evaluated the applicability of several recently commercialized SEC columns packed with 2.5 and 3 µm particles, having average pore sizes of 450 and 1000 Å. Prototype 5 µm 2500 Å particles were also tested. Moreover, two columns were coupled in series to evaluate the potential benefits of a 600 mm long separation bed, and the effect of flow rate was also studied. Different mobile phase conditions were systematically tested, using either Tris or phosphate buffer, pH 6.5 or 7.5. Similarly, we tested the effect of different co-solvents (acetonitrile or isopropanol) and additives (MgSO_4_ or MgCl_2_ included in small proportions in the mobile phase). Finally, optimized conditions were applied to a wide range of products, including 12 different mRNA samples ranging from 1000 to 4500 nt either non-modified or fully modified with 5moU. Samples from different suppliers were also tested, as well as stressed mRNA samples.

## 2. Results and Discussion

### 2.1. Evaluation of Chromatographic Conditions

To the best of our knowledge, there is limited literature on SEC analysis of mRNA. However, with the recent commercialization of SEC columns packed with particles of 1000 Å ultra-wide pore sizes by several providers, we have assessed this analytical approach as a viable method to ensure the safety and quality of the mRNA products, which are becoming increasingly popular.

A similar evaluation of SEC for the characterization of mRNA size variants has recently been published by De Vos et al. [18]. Our work differs from this previous study, as we employed low adsorption columns from a different supplier to validate and extend the previous findings. In addition, De Vos et al. have shown that elevated temperatures of the mobile phase can be beneficial to increase the resolution when analyzing mRNA. In contrast, we propose specific additives in the mobile phase to improve resolution. Finally, our optimal SEC conditions were applied to a wide range of mRNA products to assess its broader applicability.

#### 2.1.1. Optimal Pore Size for mRNA Analysis

One of the most important parameters in SEC is the pore size and pore size distribution of the packing material, as it primarily determines the selectivity of the separation. SEC columns with pore sizes of 200–300 Å are classically used for the analysis of mAb size variants, as the size of the monomeric species is often around 5 nm [19]. In the case of mRNA, a sample with 1929 nt has been reported to have a hydrodynamic diameter of 43 nm [20], so SEC columns with larger pore sizes are required.

Here, three different packing materials were tested. All of them are chemically similar in being hydrophilically modified, but the particle sizes (2.5, 3 and 5 µm), their nominal pore sizes (450, 1000, and 2500 Å), and pore size distributions are different. At the time of the study, these were prototype columns. Since then, the 450 Å and 1000 Å columns have been commercially available (branded as GTxResolve™ Premier columns). Table 1 lists the experimentally measured median and mode values of the pore sizes, the pore size ranges, and the pore volumes. On these three columns, we injected an RNA ladder consisting of species ranging in size from 200 to 6000 nt. This ladder is representative of the size of real mRNA samples (1000 to 4500 nt) analyzed later in this work, and potentially some of their impurities varying in length (shortmers and longmers). Figure 1 shows the corresponding chromatograms obtained using generic mobile phase conditions (inspired by [17]). It is clear that the separation on the smallest pore size SEC column was insufficient. In fact, only six main peaks were partially separated on this column. The situation was even worse with the 2500 Å SEC column, as only four main peaks were observed. This loss of performance can be attributed to the inappropriate pore size, but also to the fact that this column was packed with larger 5 µm particles, resulting in a reduced number of theoretical plates. Finally, the best results were obtained with the column packed with 3 µm particles and a mean pore size of 1000 Å. Eight expected peaks were observed with the RNA ladder even if the species were not baseline resolved. These results confirmed the previous findings of Goyon et al. [17] and De Vos et al. [18], who used SEC columns with pore sizes of 1300 Å and 1000 Å, respectively.

#### 2.1.2. Influence of Buffer and Mobile Phase pH

After selecting the column with suitable pore size, we focused our attention on the composition of the mobile phase. When analyzing mAbs, the mobile phase is often composed of 50 mM phosphate buffer and 200 mM KCl at neutral pH, to limit possible ionic interactions that can occur under SEC conditions [21]. These phosphate-based conditions (100 mM phosphate at pH 7.0) were also used in the work of De Vos et al. for the characterization of mRNA samples [18]. On the other hand, Goyon et al. used a buffer consisting of 50 mM Tris and 200 mM KCl at pH 7.5 [17]. In the present work, these two buffers were compared at 50 mM buffer strength in the presence of 200 mM KCl, using two different pH values (6.5 and 7.5). It is important to note that the Tris has a limited buffering capacity at pH 6.5 as it has a pK_a_ of 8.1. The corresponding chromatograms are shown in Appendix A for three different samples, namely, RNA ladder, EGFP mRNA (996 nt), and Cas9 mRNA (4521 nt). The effect of buffer type and pH on the separation was quite modest, but the overall performance was found to be slightly better with the Tris buffer at pH 7.5. Indeed, the resolution between 200 and 500 nt on the RNA ladder increased from 1.11 to 1.17 between the Tris at pH 6.5 and pH 7.5, while the resolution between 4000 and 6000 nt increases from 1.12 to 1.18. On the other hand, the resolution was also lower with the phosphate buffer for both peak pairs at pH 7.5 (resolution of 1.13 and 1.08 for the 200/500 nt and 4000/6000 nt, respectively).

In addition to the RNA ladder, Appendix A shows the chromatograms obtained for two representative mRNA samples. In this case, resolution cannot be accurately estimated by the software nor by calculation as it was too low, but visual inspection of the chromatograms confirms that the separation of EGFP mRNA was poorer with the phosphate buffer at pH 6.5, while the difference between the three other conditions was negligible. For the Cas9 mRNA size variants, identical separation was obtained, regardless of the buffer type and pH.

These results confirm that minor differences were observed between Tris and phosphate buffer in the pH range of 6.5 to 7.5 and that all conditions within this range can be successfully used for mRNA characterization. This result was expected because the phosphate groups have a very low pK_a_ value (around 2.1), and the nucleobases have pK_a_ values above 9, so the influence of the pH change so far from the pK_a_ of the molecules has obviously a limited effect on the SEC separation. However, slightly better performance was obtained using 50 mM Tris buffer at pH 7.5 in the presence of 200 mM KCl. This subtle effect might be a result of altering the zeta potential of the packing material.

#### 2.1.3. Use of Additives in the Mobile Phase

The main idea behind the use of mobile phase additives was to reduce any potential adsorptive interactions under SEC conditions. The interaction of nucleic acids with SEC packing materials has not yet been studied in detail. Hydrogen bonding might play a role in non-ideal analyte behavior, since mRNA molecules can form extensive hydrogen bonding interactions. In this context, several types of additives were tested. Organic solvents are used in SEC to reduce hydrophobic secondary interactions. Here, alcohols were first tested because they, like water, can still participate in bulk mobile phase to analyte interactions. Accordingly, 5 and 10% isopropanol were added to the mobile phase, and the corresponding results are reported in Appendix A. As shown, a decrease in resolution was observed on the RNA ladder, for the 1000/1500 nt peak pair (see arrow in Appendix A), especially at 10% isopropanol. On the other hand, no change in resolution was observed for the two representative mRNA samples. Thus, the effect of isopropanol was found to be negligible. This observation about the compatibility of isopropanol containing mobile phase is noteworthy. Indeed, a mobile phase containing isopropanol could enable the analysis of unpurified mRNA samples as they could contain residual lipids likely to adsorb on the stationary phase under aqueous conditions. In addition, it could allow the analysis of LNP mRNA samples disrupted with a surfactant such as Triton X-100 [22].

In another investigation, we tested the effects of salts containing multivalent ions, such as magnesium ions [23]. It has previously been reported that multivalent ions (such as magnesium) can have a significant effect on nucleic acid structure (folding) [24,25]. For this reason, MgSO_4_ and MgCl_2_ were added to the mobile phase in small amounts (2 and 10 mM) as a higher concentration of Mg^2+^ ions is known to impact negatively the structure of nucleic acids as reported elsewhere [24,25]. The corresponding chromatograms are shown in Figure 2 and Appendix A. For the RNA ladder, an overall reduction in resolution was observed in the presence of MgSO_4_ and MgCl_2_, compared to the reference chromatogram without addition of magnesium salts, and this was particularly critical for the 1000–1500 nt peak pair (see arrows in Figure 2 and Appendix A). This behavior was expected as RNA ladders are not representative samples to reflect RNA folding nature. For the EGFP mRNA, the shoulder is still present on the main peak regardless of the amount of MgSO_4_, while it coalesced with the main peak at 10 mM MgCl_2_. Interestingly, the separation of Cas9 mRNA high-molecular-weight species (HMWS) and low-molecular-weight species (LMWS) was improved in the presence of MgSO_4_ and MgCl_2_ salts. This improvement in separation was particularly evident at the highest levels of MgSO_4_ and MgCl_2_. Thanks to this better separation, the amount of HMWS increased from about 5% (0–2 mM MgSO_4_ and MgCl_2_) to about 9% with 10 mM MgSO_4_ and 12% with 10 mM MgCl_2_. In conclusion, it appears that MgCl_2_ may be a valuable additive for improving the separation of some mRNA samples. It might be that magnesium ion effects are sequence and/or size dependent and impact more or less the folding of the mRNA sample [24,25].

Chaotropic agents, such as perchlorate ions, also tend to disrupt possible hydrogen bonds, leading to non-ideal analyte behavior in SEC, and they may also have an impact on the tertiary structure of the mRNA. Therefore, 2 and 10 mM sodium perchlorate wereadded to the mobile phase. No apparent effect was detected at the tested concentrations, on the resolution of the mRNA ladder, EGFP mRNA, and Cas9 mRNA, so this strategy was not considered further.

Finally, we also evaluated the effect of acetonitrile (5 and 10%) in the mobile phase, as it is aprotic and therefore should behave quite differently from isopropanol. It also introduces π electrons into the mobile phase in a way that would alter any potential base stacking effects. Nevertheless, its effect on the separation of mRNA samples was also found to be insignificant as illustrated in Appendix A.

In summary, the most promising additive for SEC of mRNA uncovered by these studies has seemed to be Mg^2+^ ions, which were interesting when analyzing the Cas9 mRNA sample.

#### 2.1.4. Tuning Column Length and Mobile Phase Flow Rate in SEC

Another strategy to improve chromatographic resolution in SEC is to increase the column length or to decrease the mobile phase flow rate. The number of plates for a separation should be directly proportional to the column length in SEC. Meanwhile, working at an increasingly low linear velocity will minimize mass-transfer-related band broadening. These two solutions were tested and compared, and the corresponding results are shown in Figure 3. In Figure 3B,C, a better resolution between HMWS and the main peak was obtained compared to the initial conditions (Figure 3A), but the resolution cannot be calculated.

The effect of increasing the column length and decreasing the mobile phase flow rate was found to be identical for the separation of size variants of mRNA, so we decided to keep the column length at 30 cm and decrease the flow rate to 0.05 mL/min. This was the best solution, because it reduces the cost of materials, and the performance is similar. In addition, there is less risk of potential loss of mRNA sample on the multiple frits present when the two columns are coupled in series. Of course, the combination of two SEC columns coupled in series and reducing the flow rate to 0.05 mL/min could be an even better strategy to improve resolution, but the analysis time would be doubled, which is unacceptable for many laboratories.

Interestingly, we performed the same experiments at two different flow rates and using two columns in series, but in the absence of MgCl_2_ in the mobile phase. Under these conditions, we observed an unexpected increase in the amount of LMWS in the sample. The same behavior was observed when using only one column at a twice lower flow rate, suggesting that increased pressure from extending column length was not the factor responsible for the increased presences of fragments. To better understand this behavior, some additional experiments were performed, and they have been described in Section 2.1.5.

#### 2.1.5. Residence Time in the SEC Column

In order to understand the behavior observed in the absence of MgCl_2_, when using longer columns or lower flow rates, some additional peak parking experiments were performed, as we wanted to know if the observations could be attributed to the residence time inside the column rather than an effect resulting from increasing the column length (migration distance) or decreasing the flow rate.

The experiments were performed both in the presence and in the absence of 10 mM MgCl_2_, with a single column of 30 cm and a flow rate of 0.1 mL/min. These conditions are considered as the reference and do not generate an increase in the amount of LMWS.

A series of experiments were performed in which the flow was stopped 10 min after the injection was performed, allowing some time for the mRNA sample to stay in the column for a longer period of time. The flow was stopped for 10, 20, 40, and 60 min. The corresponding results are presented in Figure 4. As shown, the chromatographic profiles are quite different in the absence and presence of MgCl_2_. The chromatographic profiles were very consistent in the presence of MgCl_2_, while significant differences were observed in the absence of MgCl_2_, with an increasing amount of LMWS over time.

In the presence of MgCl_2_, the amount of HMWS varies between 9 and 11% between 0 and 60 min stop time, while the amount of LMWS varies between 20 and 17%. These values are quite stable, and the small differences observed are probably due to the integration of these peaks, related to the limited separation power of SEC for this mRNA sample.

On the other hand, the variations observed in the absence of MgCl_2_ were much higher, especially for LMWS, with values ranging from 25 to 43% between 0 and 60 min stop time. The variation of %HMWS was moderate with values ranging from about 5 to 2% from 0 to 60 min stop time. Upon experiencing long column residence times, it is possible that the mRNA sample began to adsorb to the packing material through various heterogeneous modes of interaction. It is also probable that these experiments are bringing to light some important considerations regarding mRNA analyte stability. Unmodified, single-stranded RNA can be rapidly degraded by contaminating nuclease activity. Nucleases might come to contaminate mobile phases, LC instruments, and columns, unless special care is taken. In this study, MgCl_2_ might have been beneficial to the analysis by altering the contaminating nuclease activity or structurally stabilizing the mRNA analyte itself (folding), thus making it more resistant to degradation [26,27].

### 2.2. Application to the Analysis of mRNA Samples

#### 2.2.1. Analysis of a Wide Range of mRNA Samples

Finally, the optimized SEC method was applied to a wide range of 12 different mRNA samples, ranging in sizes from 996 and 4521 nt, and in some cases including base modifications (5moU). The chromatograms were obtained using the 300 × 4.6 mm, 3 µm, 1000 Å SEC column, with a mobile phase consisting of 50 mM Tris and 200 mM KCl at pH 7.5, as well as a flow rate of 0.05 mL/min, resulting in maximum analysis times of about 70 min. All the chromatograms were acquired in the presence or in absence of MgCl_2_ to better understand when MgCl_2_ was beneficial in terms of performance and if there was a correlation with any specific mRNA properties (size, sequence, modifications, etc.). Because of the limited amount of sample available and the high cost, no replicate analyses were performed for these experiments. Figure 5 shows the corresponding chromatograms and proves the successful application of optimized SEC conditions for different mRNA samples. In Figure 5, blue traces were obtained in the absence of MgCl_2_, while green traces were produced in the presence of MgCl_2_. A first observation is that in the presence of MgCl_2_, the total peak area of mRNA species was systematically reduced by 10 to 30% depending on the sample, and the elution time increased by 3 to 10%. These two findings might be linked to the folding of the single-stranded nucleic acid in the presence of MgCl_2_, as noted earlier [24,25]. The folding process reduces the mRNA molecule hydrodynamic radius, which could account for later elution in SEC analysis. Furthermore, this folding could contribute to the decreased peak intensity since the nucleobases (the UV-active components of the molecule) are likely differently arranged upon the conformation change, affecting their absorption characteristics [28]. Alternatively, in the absence of magnesium ions, electrostatic repulsion may occur between the negatively charged mRNA molecules and the residual silanol groups on the packing material surface. This interaction could result in improved recovery and, therefore, heightened sensitivity. Conversely, when magnesium ions bind to the phosphate groups within the mRNA structure, electrostatic repulsion is hindered, potentially allowing for adsorption events and consequential sensitivity loss to occur.

Secondly, resolution between mRNA, HMWS, and LMWS was also modified in the presence of MgCl_2_, as shown in Figure 5, and three different situations were observed. In some cases, the chromatographic resolutions for a few mRNA samples were identical in the presence and absence of MgCl_2_. This was the case for Renilla Luc mRNA 5moU, Cre mRNA 5moU, mCherry mRNA 5moU, and FLuc mRNA 5moU. For three other samples, including EGFP mRNA, beta gal mRNA, and Cas9 Nickase mRNA 5moU, a slight decrease in resolution was observed in the presence of MgCl_2_. Finally, for the remaining five samples, namely, Cas9 mRNA, EGFP mRNA 5moU, FLuc mRNA, Cas9 mRNA 5moU, and beta gal mRNA 5moU, an improvement in resolution was observed in the presence of MgCl_2_. Of all the mRNA samples tested, the beneficial effect of MgCl_2_ was particularly evident for Cas9 mRNA and FLuc mRNA, with a significant improvement in HMWS separation.

Finally, in addition to the change in resolution, a variation in the amount of LMWS and HMWS was also observed when MgCl_2_ was added to the mobile phase, as shown in Figure 6. However, these results indicating the %LMWS and %HMWS should be interpreted with caution, as only a single chromatographic measurement was performed for each mRNA sample. For interested readers, Appendix A provides the start and stop integration times for the LMWS and HMWS for all mRNA samples. In the most extreme cases, the amount of LMWS or HMWS can changed by a factor of 2 in the presence vs. absence of MgCl_2_. In all cases, the amount of LMWS was systematically increased in the absence of MgCl_2_, while the amount of HMWS was increased with MgCl_2_, confirming the potential presence of nonspecific interactions and the importance of mobile phase optimization for accurate quantification of LMWS and HMWS. This modification could be attributed to a change in the quality of the separation (resolution), making the integration of HMWS and LMWS zones more or less difficult, but also to the behavior previously reported in Section 2.1.5., which makes that the %LMWS and %HMWS might be more reliable in presence of MgCl_2_.

Unfortunately, it was not possible to establish a direct relationship between the mRNA properties and the chromatographic behavior in the presence or absence of MgCl_2_. In fact, it was not necessarily the largest mRNA or the 5moU-modified samples that showed the most significant advantages in the presence of MgCl_2_. These observations are consistent with our hypothesis that magnesium ions might interact with mRNA in a sequence-specific manner. Based on these different examples, it is clear that MgCl_2_ should be tested during method development as a possible way to improve performance in SEC, despite the occasional absence of overall performance benefits.

#### 2.2.2. Is SEC a Stability-Indicating Method for mRNA?

Next, two different samples, namely, EGFP mRNA and Cas9 mRNA, were heat stressed at 80 °C for 10 min, or at 40 °C for 4 h, and the results were compared with the unstressed samples. The corresponding SEC chromatograms obtained, using MgCl_2_ as a mobile phase additive, are shown in Figure 7, and the amounts of LMWS and HMWS are reported in Table 1. As the temperature of the Cas9 mRNA samples was increased, the amount of HMWS was decreased, while the amount of LMWS was increased. This trend was much more pronounced at 80 °C for 10 min compared to 40 °C for 4 h. These observations correspond to those previously reported by both Goyon et al. [17] and De Vos et al. [18]. On the other hand, the same behavior was also observed with the smaller mRNA sample (EGFP mRNA), but only minor differences could be seen between the amounts of HMWS and LMWS at 80 °C 10 min compared to 40 °C 4 h. These results prove that the optimized SEC method can be considered to be a stability indicating analysis.

Cas9 and EGFP mRNA samples from two different suppliers were also tested, and the results are shown in Figure 8 and Table 2. The chromatographic profiles look very different depending on the provider. In the case of EGFP mRNA, the HMWS species were much better separated from the main peak, suggesting that larger aggregates were probably present in the sample. The amount of LMWS, however, was comparable in both samples and present at very low levels. For the Cas9 mRNA, the sample from provider A contains a much higher proportion of HMWS (27% vs. 14%), and the size of these aggregates appears comparable (identical elution time in SEC). Conversely, the amount of LMWS was reduced in the sample from provider B (4% vs. 10%). This experiment clearly demonstrated the power of SEC for quality assessment of different mRNA samples. These results also suggest that the manufacturing process could have a greater contribution to the formation of aggregates in comparison to the mRNA lengths and sequences.

#### 2.2.3. Fast Analysis of mRNA Samples in SEC

In the final part of our work, we evaluated if our method is amenable to high-throughput analysis of mRNA HMWS and LMWS by adjusting column length and mobile phase flow rate. Employing the mobile phase conditions previously optimized with the inclusion of magnesium ions (50 mM Tris, 200 mM KCl, 10 mM MgCl_2_), we encountered challenges in separating LMWS and HMWS for the four selected mRNA samples when the analysis time was below 10 min, as depicted in Figure 9A. To address the compromised resolution resulting from reduction in column length and increase in flow rate, we explored the utilization of elevated temperatures in SEC. It is indeed well known that higher temperatures enhance the diffusion coefficients of mRNA molecules, thereby boosting plate count. A similar strategy was recently proposed by De Vos et al. who used temperatures reaching up to 80 °C [18]. The fast SEC separations of mRNA samples at 50 °C are presented in Figure 9B. It is evident that at 50 °C, the peaks were narrower, facilitating improved separation of mRNA species and clearer resolution of LMWS and HMWS. Peaks were sharper at 50 °C, leading to enhanced separation of mRNA species and better resolution of LMWS and HMWS. This enhancement becomes particularly significant for larger mRNA species, implying that such conditions hold promise for expediting the testing of such materials.

## 3. Materials and Methods

### 3.1. Chemical and Reagents

Type 1 water was provided by a Milli-Q™ purification system from Millipore (Burlington, MA, USA). Potassium phosphate dibasic, potassium phosphate monobasic, Tris(hydroxymethyl)aminomethane (Tris) base, hydrochloric acid (HCl), magnesium sulfate (MgSO_4_), magnesium chloride hexahydrate (MgCl_2_), sodium perchlorate (NaClO_4_), isopropanol (IPA), acetonitrile (ACN), and potassium chloride (KCl) were purchased from Sigma Aldrich (Buchs, Switzerland). The pH of the mobile phase was controlled using a SevenMulti S40 pH meter (Mettler Toledo, Greifensee, Switzerland).

RNA Ladder 200–6000 (Reference DNF-382-U020) was purchased from Agilent (Waldbronn, Germany). CleanCap™ EGFP mRNA (040L-7601-1000) and CleanCap Cas9 mRNA (040L-7606-1000) test IVT samples were purchased from Provider A at 1 mg/mL in 1 mM sodium citrate buffer (pH 6.4–6.5). These samples were further diluted to 0.1 mg/mL with RNase-free water. For the final part of the study (Section 2.2), EGFP mRNA and CleanCap Cas9 mRNA were purchased from a second provider (provider B). The series of 12 different mRNA samples, including those with modified 5-methoxyuridine (5moU) residues, were all purchased from Provider A. Stressed samples of CleanCap EGFP mRNA and CleanCap Cas9 mRNA were obtained by performing a heat treatment at 80 °C for 10 min, or at 40 °C for 4 h.

### 3.2. Instrumentation and Columns

Measurements were performed on an ACQUITY^TM^ UPLC^TM^ H-Class Bio System (Waters, Milford, MA, USA) equipped with a quaternary solvent delivery pump, an autosampler including a 15 µL flow-through needle (FTN) injector (rinsing solvent was a mixture of 85/15 water/methanol), and a tunable UV (TUV) detector equipped with a 5 mm path length titanium cell of 1500 nL volume operating at 260 nm (10 Hz data acquisition rate and 0.5 s response time). All the experiments were performed at room temperature (20 °C). Data acquisition, data processing, and instrument control were performed using Empower^TM^ Pro 3 software.

Separations were performed on different SEC columns with varying pore sizes: commercial GTxResolve™ Premier BEH™ SEC Column (150 × 4.6 mm, 450 Å, 2.5 µm), GTxResolve Premier SEC 1000 Column (150 × 4.6mm and 300 × 4.6 mm, 1000 Å, 3 µm), and an ultrawide pore prototype SEC (150 and 300 × 4.6 mm, 2500 Å, 5 µm) column (Waters Corporation). Prototype SEC columns contained silica-based packing material modified with a hydroxy PEG (OH-PEG) bonding [17].

### 3.3. SEC Chromatographic Conditions

The optimal mobile phase consisted of 50 mM Tris buffer at pH 7.5 in the presence of 200 mM KCl and 10 mM MgCl_2_. It was filtered through a 0.45 µm filter and delivered isocratically at a flow rate of 0.1 or 0.05 mL/min at ambient temperature. The injection volume was systematically set to 1 µL for all SEC-UV experiments.

## 4. Conclusions

The aim of this study was to demonstrate the applicability of SEC for the characterization of mRNA size variants. In the present work, different low-adsorption hardware columns were tested. Each column contained hydrophilically modified particles, but different particle size and pore size. A 3 µm 1000 Å packing material was tested alongside 2.5 µm 450 Å and 5 µm 2500 Å packing materials. It is clear that the 1000 Å pore size was optimal for mRNA products ranging from 1000 to 4500 nt in length. The type of buffer and its pH appeared to have a negligible effect on the SEC separation, and we also found that the addition of organic solvents (isopropanol or acetonitrile) or chaotropic agents (perchlorate ions) does not negatively affect the separation. However, the addition of 10 mM MgCl_2_ to the mobile phase was found to, in some cases, have a beneficial effect to improve chromatographic resolution and/or help preserve mRNA from confounding on-column effects. Unfortunately, we could not find a direct correlation between this improvement in performance and known characteristics of the mRNA sample, so this solution needs to be tested through continued method development studies. To further improve resolution, we also tried increasing column length (60 cm vs. 30 cm) and decreasing the mobile phase flow rate (0.05 mL/min vs. 0.1 mL/min). The same improvement in resolution was observed, so reducing the flow rate by a factor of 2 was selected as the best option. However, care should be taken when increasing the residence time. We have shown that the HMWS and LMWS quantification is much more reliable in the presence of 10 mM MgCl_2_. Finally, it is important to note that SEC can be considered as a stability indicating method for mRNA samples. This was demonstrated in this work using 12 different mRNA samples ranging in size from 996 to 4521 nt, including in some cases modifications (5moU), stressed/unstressed mRNA samples, and mRNA from different suppliers. The example separations put forth in this work should be of help to process and analytical development laboratories working on mRNA, most especially given the resolution and robustness considerations that have been outlined. Because of the paucity of literature on mRNA, we would encourage the field to continue investigating salt effects and method development studies.

## Figures and Tables

**Figure 1 ijms-25-06254-f001:**
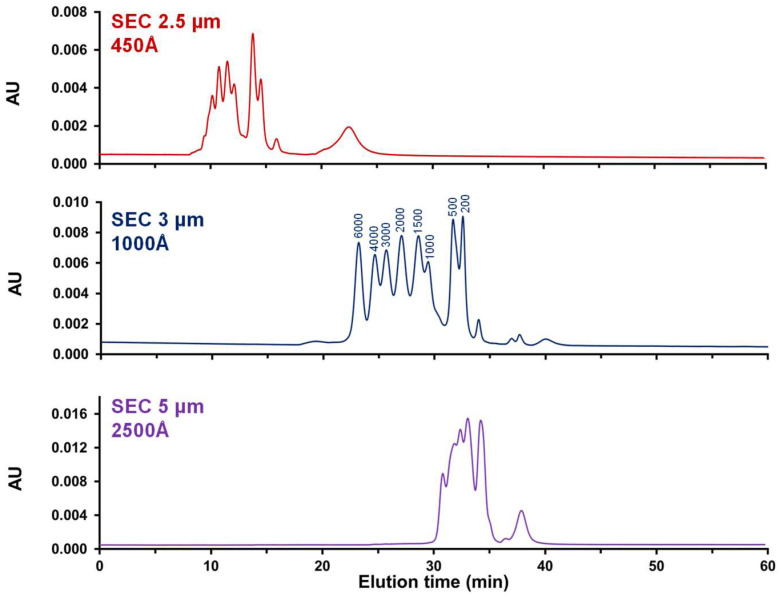
Evaluation of various packing materials with pore sizes ranging from 450 to 2500 Å for the analysis of RNA-based samples. Experiments were performed with the RNA ladder, using a mobile phase made of Tris and KCl at pH 7.5. The flow rate was 0.1 mL/min and 260 nm UV detection.

**Figure 2 ijms-25-06254-f002:**
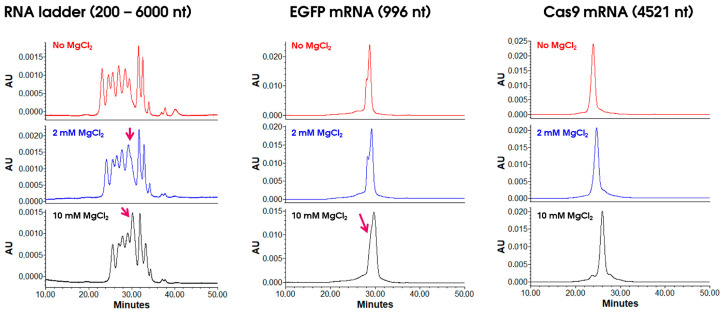
Evaluation of MgCl_2_ addition (2 and 10 mM MgCl_2_) to the mobile phase for three different samples, namely, RNA ladder, EGFP mRNA (996 nt), and Cas9 mRNA (4521 nt).

**Figure 3 ijms-25-06254-f003:**
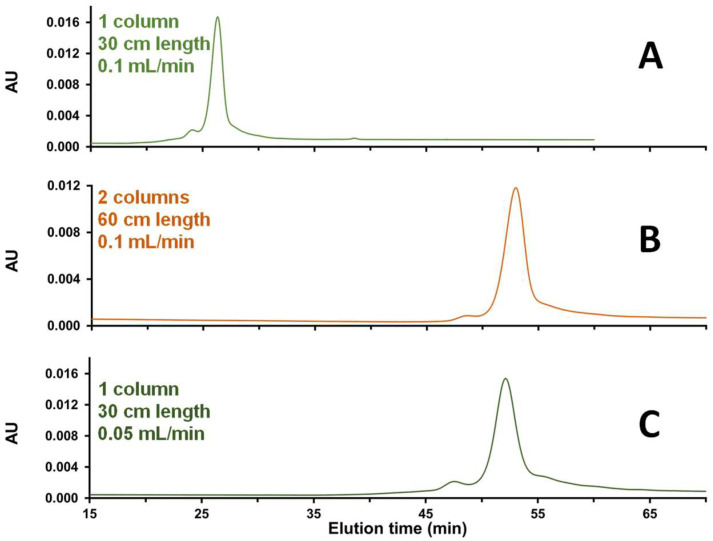
Analysis of Cas9 mRNA sample using a mobile phase consisting of 50 mM Tris, 200 mM KCl, and 10 mM MgCl_2_ at pH 7.5. In this case, different column lengths were tested: a single column of 30 cm (**A**,**C**) or two 30 cm columns coupled in series (**B**) and different flow rates: 0.1 mL/min (**A**,**B**) or 0.05 mL/min (**C**).

**Figure 4 ijms-25-06254-f004:**
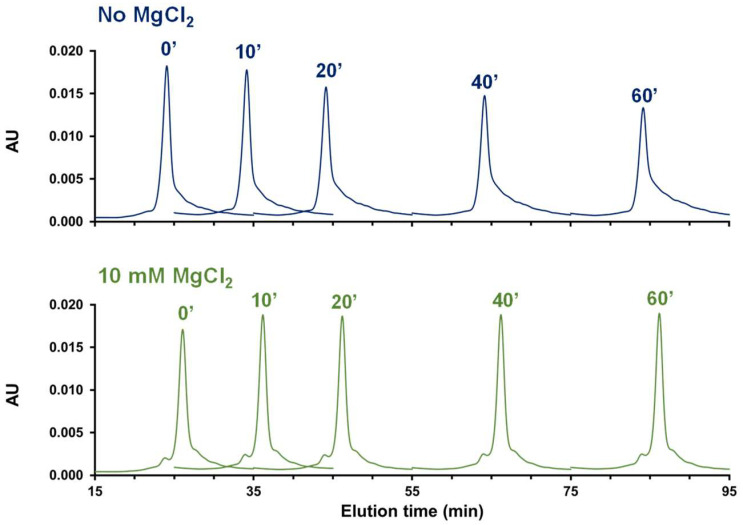
Peak parking experiments performed with Cas9 mRNA (overlay of the 30 min elution window for each chromatogram) in the absence (blue traces) and in the presence of MgCl_2_ (green traces). The values 0′, 10′, 20′, 40′, and 60′ correspond to 0, 10, 20, 40, and 60 min of flow set at 0 mL/min during the analysis (flow was stopped 10 min after the injection).

**Figure 5 ijms-25-06254-f005:**
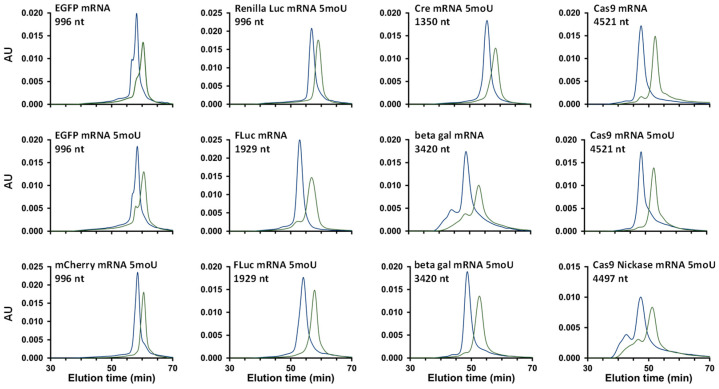
Analysis of 12 different mRNA samples on a 30 cm column with 50 mM Tris, 200 mM KCl, at a flow rate of 0.05 mL/min under two conditions differing by the presence of additive (blue traces, no additive; green traces, +10 mM MgCl_2_).

**Figure 6 ijms-25-06254-f006:**
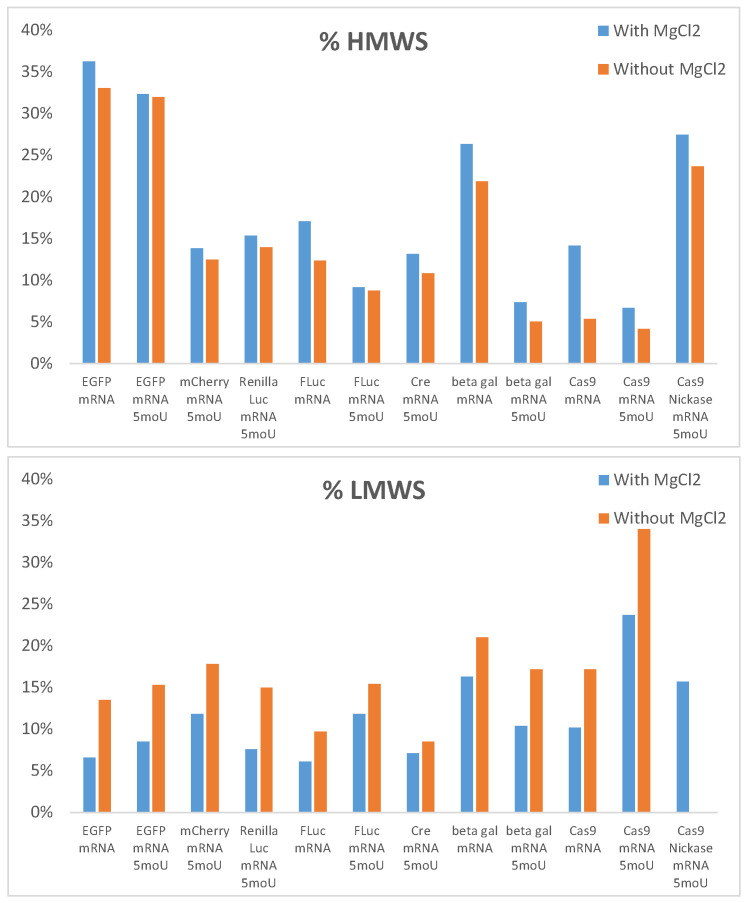
Relative amount of LMWS and HMWS (expressed in %) in several mRNA samples in presence and absence of MgCl_2_ in the mobile phase.

**Figure 7 ijms-25-06254-f007:**
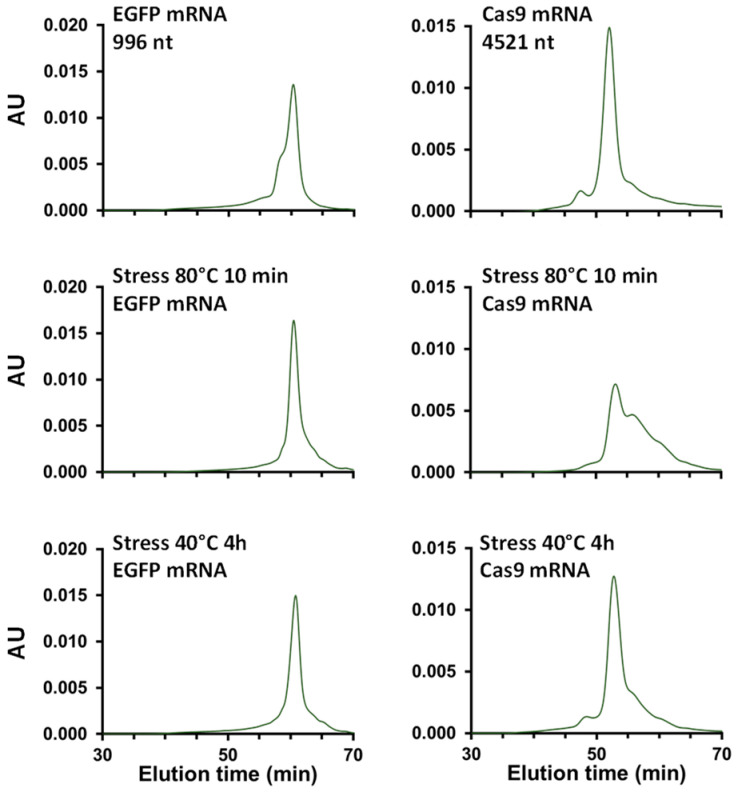
Comparison of unstressed vs. heat stressed at 80 °C for 10 min, or at 40 °C for 4 h for EGFP mRNA and Cas9 mRNA samples. The samples were analyzed on a 30 cm SEC column with 50 mM Tris, 200 mM KCl, and 10 mM MgCl_2_ at a flow rate of 0.05 mL/min.

**Figure 8 ijms-25-06254-f008:**
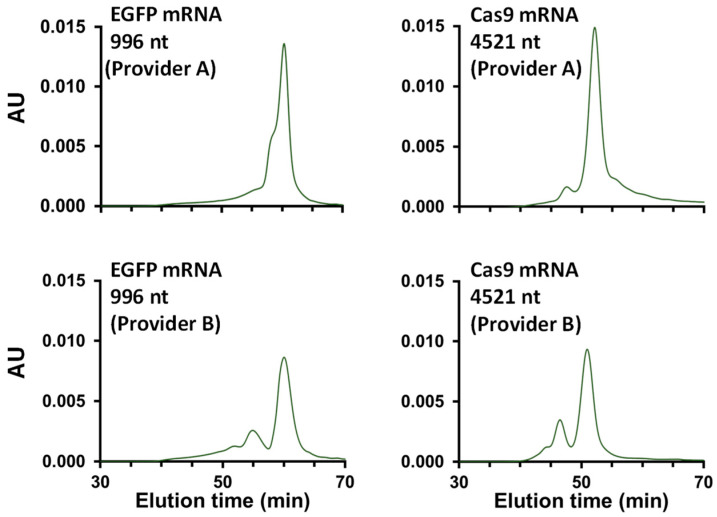
Analysis of EGFP and cas9 mRNA samples obtained from two different providers (Providers A and B). The samples were analyzed on a 30 cm SEC column with 50 mM Tris, 200 mM KCl, and 10 mM MgCl_2_ at a flow rate of 0.05 mL/min.

**Figure 9 ijms-25-06254-f009:**
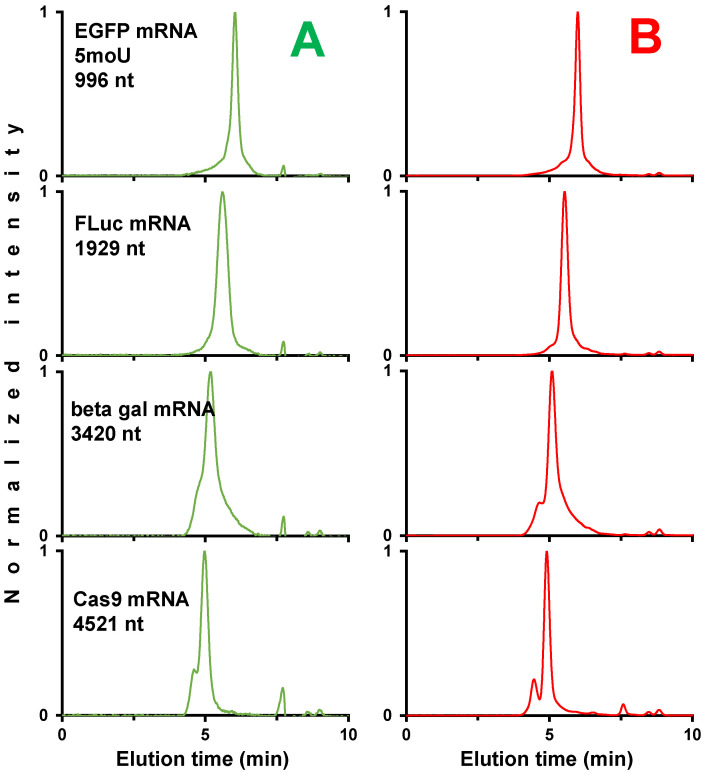
Normalized chromatograms for mRNA samples of different sizes showing fast SEC separations using 4.6 × 150 mm column and 0.25 mL/min flow rate performed at ambient temperature (**A**, green traces) and 50 °C (**B**, red traces). The mobile phase was composed of 50 mM Tris, 200 mM KCl, and 10 mM MgCl_2_.

**Table 1 ijms-25-06254-t001:** Experimentally measured pore size, pore size distribution, and pore volume data of the columns studied. The mercury intrusion porosimetry measurements were performed on a MicroActive AutoPore V 9600 system.

Nominal Pore Size (Å)	Median (Å)	Mode (Å)	Pore Size Range (Å)	Pore Volume, cc/g
450	427	453	190–650	1.07
1000	965	884	400–2000	0.79
2500	2458	2578	1500–4500	0.86

**Table 2 ijms-25-06254-t002:** Relative amount of LMWS, HMWS, and mRNA (expressed in %) in several stressed mRNA samples and mRNA samples obtained from different providers in the presence of 10 mM MgCl_2_.

Sample	% HMWS	% mRNA	% LMWS
Cas9 mRNA	14%	75%	10%
Cas9 mRNA stressed 80 °C 10 min	5%	40%	53%
Cas9 mRNA stressed 40 °C 4 h	11%	62%	26%
EGFP mRNA	36%	57%	6%
EGFP mRNA Stressed 80 °C 10 min	15%	65%	19%
EGFP mRNA Stressed 40 °C 4 h	22%	62%	15%
Cas9 mRNA Provider A	14%	75%	10%
Cas9 mRNA Provider B	27%	68%	4%
EGFP mRNA Provider A	36%	57%	6%
EGFP mRNA Provider B	36%	59%	4%

## Data Availability

Data are contained within the article and Appendix A.

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
