# Peer review of "Optimizing Messenger RNA Analysis Using Ultra-Wide Pore Size Exclusion Chromatography Columns"

_ijms, 2024, doi:10.3390/ijms25116254_

Round 1

Reviewer 1 Report

Comments and Suggestions for Authors

In the article the authors describe development of mRNA analytical platform employing size exclusion chromatography (SEC) to analyze wide range of mRNAs. The authors studied various factors including particle and pore size, buffer, pH, additives, column length, flow rate, and temperature, and identified the effects of these factors on mRNA separation.  Overall, the authors were able to test factors that could potentially improve the analytics of mRNA purification and characterization.  However, most of these factors were already studied and reported in De Vos et al. (Ref18, https://doi.org/10.1016/j.chroma.2024.464756) and the authors have obtained similar conclusion as reported pulication. Therefore, this work as such does not give significant additional input/insights into using SEC to analyze a wide range of mRNAs. A detailed analysis and study employing other factors, not studied previously, are needed to improve the significance and novelty of this study.

A few minor comments:

·       Line 76: Define/abbreviate CE.

·       Line 228: “Figures S2 and 3”, not S3.

Reviewer 2 Report

Comments and Suggestions for Authors

In the manuscript titled "Optimizing Messenger RNA Analysis using Ultra-Wide Pore Size Exclusion Chromatography Columns", D'Atri et al. evaluated several SEC columns with varying pore sizes and re-determined that a column with 1000 Å pores is optimal for the analysis of mRNA products, regardless of the size between 500 and 5000 nucleotides. The authors demonstrated that the addition of 10 mM magnesium chloride to the mobile phase returned improved chromatographic resolution and recovery of large size variants for some mRNA samples. By exploring fundamentals of SEC, they identified that the use of longer columns and/or lower flow rates can lead to an apparent increase in the amount of low molecular weight species and peak tailing. Therefore, despite beneficial for chromatographic resolution, they demonstrated that these strategies should be avoided.

Despite not highly innovative, the community will gain from the publication of such a study. It further demonstrates the applicability of SEC for the analysis of mRNAs and provides additional strategies for the analysis of this class of molecule by SEC. Therefore, such a manuscript is suitable for publication in IJMS.

I have a few minor questions and edits to suggest though:

1. Line 79 – “reversed-phase”, not “reverse-phase”.

2. Lines 144-145 – In this part, the authors described nominal values for pore size and particle size for the columns used. However, no information regarding pore size distribution was provided. Since these are “one of the most important parameters in SEC”, and as some of the authors are affiliated to the column vendor, would it be possible to disclose more information about pore size distribution, as well as information beyond nominal values for the other column features?

3. Figure 2 – Only minor differences are reported in Figure 2. Therefore, such a figure should be moved to the supplementary file. Also, please add labels describing the samples used to acquire the results depicted in each column (as done for Figure 3).

4. Folding of mRNAs seems to be critical for chromatographic resolution and stability. Therefore, I wonder if the authors tried to use prediction algorithms to better correlate chromatographic/stability behaviors with predicted structures. If yes, why isn’t this discussed in the manuscript? If not, why not?

5. Figure 5 – values described in the text are highly dependent of how chromatographic peaks were integrated. Therefore, I think the authors should add peak integration markers to the figure (or add to the supplementary file). More importantly, how many replicates were analyzed? Considering the shape of the chromatographic peaks, I wonder what the variance of the analyses is.  

6. Line 327 – “might have been”, not “might have”.

7. Line 333 – remove “was finally” from the text.

8. Line 340 – “etc.”, not “etc….”

9. Figure 7 – histograms really require error bars to be meaningful. Therefore, I recommend running replicates so significance of the differences can be better illustrated and evaluated. In fact, a lot is discussed in the manuscript regarding differences in resolution and purity measurements. All values seem to be based on only one replicate per experiment. So, I really wonder how significant those differences really are.   

10. Anytime temperature is described in the text it has space between the number and the unit (for ex, it is written 80 °C, not 80°C). Please, remove the spaces.

11. Line 445 – “De Vos”, not “De vos”.

12. Line 461 – remove space between a word and a comma.

13. Line 473 – location of Provider A was disclosed. However, this is not the case for Provider B. Please, disclose it if possible.

14. Lines 486-491 – please be consistent regarding column names.

Round 2

Reviewer 1 Report

Comments and Suggestions for Authors

In the cover letter provided the authors have shared how their work is significant to the field and how this work compares to the previously reported results.  The authors will have to add these details in the manuscript under section 2.1 as well.

Author Response

As requested by the reviewer, we have modified section 2.1 of the manuscript to add information contained in the cover letter.